# Secularity as a Point of Reference: Specific Features of a Non-Religious and Secularized Worldview in a Family across Three Generations

## Christel Gärtner

Cluster of Excellence "Religion and Politics", University of Münster, Johannisstr. 1, 48149 Münster, Germany; cgaertner@uni-muenster.de

**Abstract:** My contribution will focus on secular and non-religious worldviews and will aim to reconstruct secular relationships with the world that develop from lived values and their transmission in the family. I will try to show in detail how a non-religious habitus develops in socialization over several generations, becomes entrenched in later biographical positioning, and shapes how a person relates to the world, including their view of religion. After a brief outline of the religious field in Germany, I will concentrate on a family case whose first generation (grandparents) grew up in the GDR. This family has had no religious socialization or child baptisms for three generations and secularity has become a positive point of reference for how its members justify their own life patterns. For the members of this non-religious family, religion still becomes selectively relevant. Using concrete situations and contexts where the family has contact with religion, I will show how these encounters become a marker for drawing boundaries. In conclusion, I will follow Quack and Schuh's distinction between "indifference to religiosity" on the one hand, and "indifference to religion" on the other, and argue that indifference to religiosity, but not indifference to religion, can be clearly identified.

**Keywords:** secular worldviews; non-religious habitus; transmission of non-religion; indifference to religiosity; indifference to religion

## 1. Introduction

My contribution will focus on secular and non-religious worldviews and will aim to reconstruct secular relationships with the world that develop from lived values and their transmission in the family. I will try to show in detail how a non-religious habitus develops in socialization over several generations, becomes entrenched in later biographical positioning, and shapes how a person relates to the world, including their view of religion.[1] Even if non-belief is not a modern phenomenon, certain social conditions are required for a secular habitus to develop, one that neither requires its bearers to justify themselves and nor forces them to accept social disadvantages. To limit my definition of non-religion, I would like to adopt a relational approach and conceptualize non-religiosity in terms of its relation to the religious field (Lee 2014; Quack and Schuh 2017). I will therefore first provide a brief outline of the religious field in Germany, a field that allows for the development of a non-religious habitus (2), followed by a brief description of the methodological approach (3). I will then concentrate on the case of a family whose first generation (grandparents) grew up in the GDR (4). This family has had no religious socialization or child baptisms for three generations, and secularity has become a positive point of reference for how its members justify their own life patterns. I have chosen this case because the middle and youngest generations (father and daughters) have no primary interest in religion and display an indifference that does not force them to criticize religion or draw boundaries in relation to it. Nevertheless, their attitude is not adequately described as "indifferent" in the strict sense of disinterest in religion (Gärtner et al. 2003; Wohlrab-Sahr and Kaden 2013).

Rather, it is more accurate to characterize it as "non-religious", since religion becomes selectively relevant to them (Lee 2014). Using concrete situations and contexts where the family has contact with religion, I will show how these encounters become a marker for drawing boundaries (5). In conclusion, I will follow Quack and Schuh's distinction between "indifference to religious beliefs, practices or affiliations (indifference to religiosity)" on the one hand, and "a lack of disposition or opinion regarding the manifestation and proper place of religion in society (indifference to religion)" on the other (Quack and Schuh 2017, p. 3), and argue that indifference to religiosity, but not indifference to religion, can be clearly identified.[2]

## 2. The Changing Religious Field: Societal Conditions in Which Non-Religion Develops

Historically, Germany was a religiously mixed country with a Protestant majority and a Catholic minority (Gärtner 2019a). At the turn of the twentieth century, 98% of the population belonged to one of the two Christian churches (Hölscher 1990; Liedhegener 2012, p. 519). However, atheism emerged around 1900 as a programmatic antithesis to the churches, this becoming more widespread in northern and eastern Germany than in the western and southern parts of the country (Schmidt-Lux 2008). Even though the *Religionspatent* of 1847 explicitly allowed people to "leave the church" without having to convert to a new church, it was not until the Weimar Republic (1918–1933) that people could do so without serious consequences (Weir 2014, p. 64), with the number of non-believers (nones), a number including those not affiliated to one of the two Christian churches, rising from 0.02% to about 5% between 1920 and 1940 (Liedhegener 2012, p. 519). This constellation remained relatively stable up to the second half of the twentieth century, which is the reason that religious affiliation can be found in almost all families, at least among the oldest generations. However, the religious field then changed profoundly: while the GDR had already witnessed a process of forced secularization after the Second World War (Wohlrab-Sahr et al. 2009), West Germany also underwent a religious crisis triggered by the push towards modernization and liberalization in the 1960s, and this gradually transformed the predominantly Christian country into a secular one (McLeod 2007).

The anti-religious politics of the SED and the struggle that it waged against the churches thus met with a population that, although still consisting largely of church members, were already quite dechurchified internally, and who in a certain sense represented a suitable precondition for a crudely scientific worldview (Schmidt-Lux 2008, pp. 132–34). When the GDR was founded in 1949, 81% of the population still belonged to the Protestant church; in 1989, 25%. The proportion of Catholics decreased during the same period from 11% to about 4%; the number of those without a religion rose from 7% to almost 70% (Pollack 2009, pp. 249–50). The cohorts born in the GDR thus grew up in a predominantly anti-church and religiously distanced environment in which not having a faith was anchored in people's lifeworlds (Großbölting 2013, p. 231). This contributed to the formation of a "secularist habitus" (Wohlrab-Sahr et al. 2009) that deprecated religion and renounced transcendent meaning. What became normal was "scientific atheism",[3] and the absence of ties to a dogmatic and institutional form of religion.

But West Germany has also seen an increasing number of people leave the church and the church's dogmatic faith lose its binding power since the 1960s, and this development contributed greatly to establishing the ideal of individualization as a norm. Norms regarding family, marriage, and sexuality have shifted within a few years, meaning that, when it comes to providing meaning, secular areas such as family and work have gradually assumed greater weight than religious areas (Gärtner 2019b; Tyrell 1993). Thus, since the 1960s, the churches have lost not only their sovereignty of interpretation in society with regard to central moral questions, but also their power to create identity. This turning point has brought about a decisive change: "Not attending church, but staying away from it, has become the social norm" (Jagodzinski and Dobbelaere 1993, p. 76; my translation). A reversal of the burden of proof has taken place, meaning that people now have to justify their ties to church religion.

Because the religious field differed in the two parts of Germany from the beginning, with these differences already apparent in the first half of the 20th century and intensifying after the Second World War, this landscape is still divided in Germany. While 31.9% of the population still belonged to the Protestant church (with 28.4% infant baptisms) and 36.5% to the Catholic church (with 30.4% infant baptisms) in West Germany in 2010 (Pollack and Rosta 2017, pp. 76–78), the percentage of church members in East Germany was much lower: 19.5% and 3.7%, respectively (Pollack and Müller 2011, p. 131).[4] We also find large differences among the non-religious: while 65% of East German respondents never belonged to a religion, 73% of West German respondents were once members of the Protestant church (EKD, p. 16).[5]

Despite these differences in the religious field between the two parts of Germany, we can also observe the steady decline of institutional religiosity and religious practice in West Germany since the 1960s (Pollack and Rosta 2017). This decline, which we examine in detail in our research project, is mainly ascribed to an intergenerational change (Stolz et al. 2016; Voas and Doebler 2011).[6] Against this background, there have been generation-specific upheavals with regard to forms of religious habitus and worldviews (Wohlrab-Sahr 2002). We can observe that it is the generations born since 1970 at the latest (earlier in the GDR), those experiencing adolescence from the mid-1980s onwards, who position themselves in a social context that is primarily secular (Gärtner 2016). For them, secularity has become the dominant perspective, meaning that being non-religious no longer requires explanation or justification, and is just as much a private matter as being religious. Nevertheless, church and religion are still part of social reality and people encounter these institutions in certain situations—be it at school through religious education, public media (Bullivant 2012), or (especially in West Germany) through family and friends at religious rituals such as baptisms, weddings, and funerals. This also applies to individuals growing up in a family where religion has no relevance either in everyday life or for how they relate to the world, since coming into contact with religion causes them, at least in part, to take a position with regard to religion (Lee 2014).

## 3. Methodological Approach

My data are drawn from the research project mentioned above. We conducted both a quantitative survey with 8400 responses as well as in-depth interviews with 15–20 families in each of the five countries involved (Canada, Finland, Germany, Hungary, and Italy) representing three different generations.

The families for the qualitative part were selected according to contrasting criteria based on the religious field as well as on different religious or non-religious traditions, with the aim being to give our sample a maximum level of variation. Most of the family interviews involved members from three generations. We opened the interviews with the same input: "We are interested in how you have passed on from on generation to the next values, beliefs, and what is important to you in general, and in what has changed in the process over time. Can you tell us what it is like in your family?" Subsequent to the interview, we collected demographic data over five generations (such as births, educational trajectories, career decisions, marriage, divorce, migration, deaths, religious orientation and change, and special life events) to create a genogram (family tree) for each family (Hildenbrand 1999, 2007; Bengtson et al. 2013, p. 14). We transcribed and anonymized the interviews before analyzing them using the method of Objective Hermeneutics (Oevermann 2000), which is a line-by-line analysis. By analyzing the opening sequence and other central sequences, we established a hypothesis about the transmission of (non-)religion and values. We falsified, confirmed, or developed in more detail the hypotheses by help of the interview text and established a case study.

The German sample consists of 14 families so far. We conducted five family interviews face-to-face and nine via Zoom. In terms of family generations (grandparents = G1, parents = G2, and children = G3), twelve family interviews involved representatives from all three generations; in two families, the children (G3) were born after 2014 and were therefore too

young to participate actively in the interview. Altogether, the interviews involved a total of 18 (13 ♀, 5 ♂) members from G1 (born 1931–1956), 27 (16 ♀, 11 ♂) from G2 (born 1951–1984), and 23 (11 ♀, 12 ♂) from G3 (born 1981–2010).[7] To analyze the different religious paths taken by siblings, we also conducted three individual interviews with brothers and sisters from the middle generation.

Seven families are predominantly Protestants, with one family belonging to the evangelical spectrum and one having broken off church ties in G1. Three families are Catholics, although in two families the G2/G3 members are each married to a Protestant spouse, with G3/G4 being baptized Catholic. One family has a long Muslim tradition and a migration background. We classified three families as non-religious, although we did so in a limited way, since all three families saw turns to religion in adolescence or adulthood. Our criterion was that the transmission of religion from G1 to G2 was interrupted, that is, in at least one family branch, neither infant baptism nor religious socialization took place.

## 4. The Case of a Non-Religious Family

*4.1. Conditions of Socialization and Their Role in the Emergence of a Secular Habitus*

This article is based on a family interview involving the paternal grandparents (G1_GM, *1951; G1_GF, *1944), their son (G2_F, *1971), and the two granddaughters (G3_D1, *2000 and G3_D2, *2004).[8] In the family conversation, the individual family members reflected on their memories of coming into contact with religion and faith, using these memories as an opportunity to discuss religion and faith and to explain their own religiosity or non-religiosity.

The absence of religious socialization already began with the first generation of the family (this does not apply to the absent daughter-in-law, G2_M, *1975, and her family of origin). Chiefly responsible for passing on non-religiosity, both paternal grandparents grew up in the GDR,[9] where they were neither baptized nor came into contact with religion at home or school.[10] As teenagers, they fled with their parents to West Germany in the early 1960s; only the paternal great-grandfather, as a Baptist, had religious ties, although his long captivity during and after the war meant that his religiosity had no influence on the upbringing of his son, the present grandfather. Nonetheless, the great-grandfather's religious ties were a reason for him to leave East Germany, especially since, being a teacher, he felt caught in a conflict of worldviews (and his two sisters were already living in West Germany). Despite his Baptist father, the grandfather did not develop any religious bonds in West Germany, either, which he explains by pointing to his closer ties to his non-religious mother. However, he tells of an experience with religion that impressed and moved him: namely, he witnessed the comforting power of faith with his aunt at the death of her beloved sister.[11]

In contrast, the grandmother, who was about 10 when she fled with her family to West Germany, met a pastor in the refugee shelter who was very caring and whose Christian attitude was so affecting that she had herself baptized and confirmed (G1_GM: 215–242). Her confirmation, which in the mid-1960s was still a self-evident and unquestioned rite of passage in West Germany, was also an opportunity for her to integrate and become a part of society. However, it was her actual experience of charity that convinced her and had a lasting impact on her further life: while not developing ties to the church or institutional religion, she is committed to what she sees as being the essence of Christianity—charity and gratitude.[12] Her commitment to these values, which she also passes on, is expressed in her respectful treatment of people and her commitment to others. She also says that she asks for support when there are difficulties or difficult decisions to be made in the family and then gives thanks, both the request and the thanks being directed "upwards". This wording can be interpreted as implying that, on the one hand, she believes there to be a transcendence, one that nevertheless remains abstract for her, and on the other, that she has internalized the idea of the ungraspable nature of life.[13] She and her husband married in 1970, when, despite the intervention of the grandfather's Baptist father, they decided against a church wedding because "we don't live this faith in any way" (G1_GM: 237).

Since society had begun its process of liberalization in the mid-1960s, there was no need to justify or explain this decision in the West Germany of the time. Additionally, when their son was born in 1971, they did not have him baptized.

The beginning of the interview sees the son reconstructing when and how he came into contact with religion ("when I was confronted with faith in the family for the first time", G2_F: 12–13). He identifies the first experience of difference as: "so I know that my grandfather [my father's father] ... I think would have described himself as ... religious" (G2_F: 14–16).[14] He recalls his grandfather going regularly to church and reading the Bible (which made it clear to him as a child: "my granddad believes in God"). In doing so, the grandfather was following a practice that differed from the practice followed in his own family, which neither practiced a religion nor believed in God. However, this experience of difference did not lead him to conclude that there is or must be a God. Since the grandfather's faith neither inspired him as a child to think about God nor encouraged him to emulate his grandfather, because he could not see how his life could benefit from faith, we suspect that the son's relationship with his grandfather was not particularly loving or warm.[15]

The second memory concerns a friend who tells him that, when he wakes up at night and goes into the kitchen, there is a flame burning on the cooker that goes out the moment he enters the kitchen. While his friend interprets this experience as a sign and thus as proof of something transcendent or of a higher power, the father deems this interpretation to be "nonsense" and states: "I've never had the feeling that there was something somewhere" (G2_F: 28). This expresses that he considers a world beyond empirical reality to be implausible. Nevertheless, he does ask himself why he has "no antenna at all" (G2_F: 26–27) for a higher power and can do nothing with it, a question that sets in motion a process of reflection. He attributes the lack of such an antenna to his socialization and to his parents, whom he addresses directly in the interview thus: "You never taught me that it was important to go to church, but I always had the feeling that what was important to you, that you also taught me . . . that you treat each other respectfully, that ... you always had such balancing thoughts and that gave me enormous security, ... I was never confronted with disputes or with fears of loss" (G2_F: 40–51). According to his reflections, this reliability and security enabled him to grow up without fear and without needing a higher power: "I always had the feeling that the world was as safe as it was" (G2_F: 53).

Although not baptized, he, like everyone else in his class, was confirmed at the age of 14. How did this decision come about? It was preceded by another experience of difference or exclusion.[16] Unlike everyone else in his class, he had not received a "letter" which caused him to panic, the reason being (as it turned out) that he had not been baptized. Since he did not want to be an outsider, he had himself baptized: "and I was confirmed because everyone in the class was confirmed" (G2_F: 54–55). This desire not to be excluded from an experience that his friends had did not cause any conflict in the family, and nor did it raise the question of why he wanted to be baptized. Baptism was simply seen as a prerequisite for him to be confirmed together with his friends. He was thus following up in a way on an experience that his mother had, whereby the unifying factor was inclusion in the (peer) community. The fact that his parents granted him this wish points to the fact that, although not religious, they also did not hold a secularist or atheist position that would have caused them to draw boundaries with regard to religion and church. Rather, they took a pragmatic stance without demanding any consequences from their son. While they encouraged him to read the Bible and to go to church after confirmation without exerting pressure or moralizing, he did so, but very quickly lost interest because he felt that neither confirmation nor the church services enriched him: "I never had the feeling that I personally lacked something in my . . . meaning of life or something" (G2_F: 73–74). Rather, he found religious service to be extremely boring: "sitting around was exhausting and I didn't understand what was being said at the front anyway" (G2_F: 75–76). On the one hand, he is talking about an experience shared by many members of his generation:

the church neither gives satisfactory answers to the existential questions that preoccupy adolescents, and nor does it speak a language that is understood and resonates, which is why the majority of young people turn their backs on the church after confirmation (Schweitzer 2017). On the other, this formulation also shows that, for him, it was enough to have a secular answer to the question of the "meaning of life".

He ends his evaluation of the biographical phase of his childhood here, noting that his "attitude towards religion" would intensify again at a later point in his life. He is certain that he lacks a "religious gene" but can imagine that many people find comfort in religion, e.g., when a close relative dies. He would deal out of interest with religion again and again in the further course of his life, and believes that "people always ascribe values to religion ... like charity" (G2_F: 93–94). Unlike his mother, he distances himself from religion in this respect because he does not consider "charity" to be necessary, "but I am friendly to everyone ... and I think there are just as nice strict believers ... as atheists ... and non-believers" (G2_F: 94–96). Even though he draws a boundary at that point, he does not have to make sweeping judgments about religious people.[17] It is important for understanding his development that his early encounters with religion (church services, religious education, and confirmation classes) did not enrich him intellectually. It was only through engaging with criticisms of religion (by Nietzsche and Feuerbach, for example) that he found the subject exciting. This is the first indication that the meaning of religion was most likely to open up for him, if at all, through an intellectually stimulating or scientific approach.

We can state first of all that, since childhood, he has shown, besides his secular orientation, a strong intellectual interest that would lead him to later turn to the field of science in his studies and professionally. This leads to the thesis that science is the guiding medium through which he perceives and classifies the world. Our analyses show that this approach is influenced by the grandfather, who establishes in the family a scientific habitus already acquired through his GDR socialization; as we shall see, this habitus will prevail into the third generation. We assume that this habitus was also able to develop through the attitude of his parents, who tried to "understand the other side" and not to impose a "higher truth on anything" (G2_F: 540–541). They also did this when it came to people (such as alcoholics, murderers, suicide cases, and rapists) whose actions could not be justified but could nevertheless be understood: "there is always a reason why someone is like that and ... this secular questioning of the reason has never somehow made me bring into play something transcendent" (G2_F: 550–551). His parents not only convey this attitude to him, but also pass it on through actual practice in the family: respectful interaction with each other, as well as the attitude of looking at things from different sides instead of through the lens of emotional involvement. All family members share this attitude of dealing rationally with problems and crises, and reject moralizing and the condemnation of others—these points are then also the starting point for criticizing established religions, which I will return to below.

Before addressing the significance of science for the father, but also for the daughters, and the drawing of boundaries involved, I would like to look once again at the encounters with religion of the individual family members who encourage an engagement with religion.

### 4.2. Encounters with Religion and Church

The grandmother came into contact with religion as a child or adolescent through a pastor who was able to convey the attitude of Christian charity to her during a time of crisis following her escape from the GDR (early to mid-1960s). He thus became a significant person in her life, someone through whom she developed her understanding of religion. This encounter led to a lasting and positive anchoring of a faith that she describes in the interview as a "childlike faith" (G1_GM: 302). Two readings can be derived from this: namely, her faith is anchored more in emotions than in ideas, and, since its source does not lie in a lived religious or churchly practice, is in a sense unencumbered. The fact that she cannot simply give up this faith makes her wonder what it is based on—whether her

mother[18] might indeed have prayed with her when she was a small child, or whether the foundations of her faith were laid during confirmation classes. On the basis of these experiences, which she herself cannot completely fathom, she considers praying to be "natural", something where she addresses questions and gives thanks "towards God" (G1_GM: 303–304). We can deduce from this that the grandmother has an understanding that not everything lies in one's own hands and that decisions can also turn out differently than intended—in this respect, her faith can best be interpreted as a sense of the ungraspable: that there is something that goes beyond everyday life.[19] This connection to transcendence distinguishes her from the rest of the family. Even though she does not consider her faith to be religion, she would miss something if she gave it up despite her criticism of the church, to which she has no close ties. While "charity and gratitude" (G1_GM: 589) are the essence of Christianity or religion for the grandmother, one of the granddaughters (G3_D1: 593) emphasizes that this is also possible without religion. The father and daughters agree that ethics is not limited to and dependent on religion.

The father's experiences in the 1970s and 1980s differ from those of the grandmother: he did not meet someone who became important for him like the pastor for his mother. He was convinced neither by his grandfather and nor by the friend whose esoteric interpretations were too abstruse for him. This was also true for the representatives of the church: confirmation and church services were more of a burden than an inspiration. He found a way to approach religion through his intellectual interest in it.

Born in the early 2000s, the daughters, like the father, grew up in a secure and comfortable environment[20]—they also have no reason to look for meaning outside their actual world. Unlike their father, though, they "actually have quite nice memories of the church" (G3_D1: 108): through, for example, religious celebrations such as baptisms, weddings, and Christmas services in the maternal grandfather's Catholic family, and also through school in England, where a school choir sings in church services and where they can enjoy the ceremonial atmosphere.[21] The older of the two sisters is also positive with regard to the "room used for confirmation" that her friends have "where people meet ..., that this is really a place where people like to come, experience things together, make things, bake biscuits, and so on" (G3_D1: 117–119). She finds this room so inspiring that she would probably decide to be baptized and confirmed if she belonged to the same congregation as her friends. Not only because this opportunity is missing, but also because they do not believe, both sisters decide against confirmation.[22] Although both sisters have many friends who have been baptized, confirmed, or who have received first communion, they suspect that very few believe in God, but rather believe "that there is something greater, that there is something that is above everything" (G3_D1: 499). This narrative demonstrates that Christian identity is not very strong among the young generation (Gärtner 2013; EKD 2014). Both distance themselves when the motive of their friends for being baptized or confirmed is not faith but the desire to receive gifts or because they want to celebrate their wedding in church—they do not perceive these motives as authentic religious.[23] Summing up their own upbringing, they conclude that it is difficult to acquire a faith if one has not been socialized religiously.

Asked by the grandmother whether she misses religion, the granddaughter points to her upbringing with science: "no, because I don't have the feeling now that anything ... is missing ... because we grew up with science and something that can actually explain everything, all questions, ... that I don't have the feeling that we missed something" (G3_D1: 419–422). Without blaming anyone, she adds: "sometimes I think that I want to know more or that I think that it would be important if I knew more" (G3_D1: 422–424). This explanation shows once again how central science is in the family for how its members relate to the world; this is especially true for the father, who also establishes and passes this approach on in the family. This scientific approach to the world, which underlies how the family members understand society and religion, also becomes a marker for drawing boundaries regarding religion.

### 5. Demarcations towards Religion Based on a Secular-Scientific Habitus

As shown by the reconstruction of this case, a habitus oriented towards science as the guiding medium for understanding the world was acquired in family socialization and then passed on in all three generations. This secular habitus is not in itself hostile to religion. However, I would like to examine how the associated preference for rational over religious interpretations of the world leads to the drawing of boundaries in relation to (lived and institutionalized) religion when it comes to explaining the world (Section 5.1), as well as to questions of ethics (Section 5.2).

*5.1. Secular vs. Religious Explanation of the World*

It was not until he was studying biology that the father engaged more intensively with religion: convinced of the theory of evolution as a secular explanation of the world, he was confronted with creationism and with people who believed in creation theology. This inspired him to engage with religion. The fact that people rejected the theory of evolution "was also an initial spark for me because I didn't understand what exactly it is that people are rejecting" (G2_F: 913–914). The fact that this encounter triggered a conflict for him makes itself clear in the language that he uses: in his studies, he had "run into contact with a few people" (a mix of "come into contact with" and "run into conflict") who, without presenting arguments, rejected the theory of evolution and instead believed (without scientific evidence) in the myth of creation (G2_F: 919). He then began to engage with the Christian religion and read the Bible, which led him to see the Bible and religion as irrational and contradictory. As evidence, he points to statements made by a boy from his Catholic family-in-law who had questioned the creation: "it all came about through the big bang and evolution" (G2_F: 870–871). He mentions also that pupils were already doubting the truth of Biblical stories even in the primary school where his wife teaches religion. He quotes them: "Jonah and the whale ... that's not true at all" (G2_F: 1007–1008). His judgments show that he does not read the Bible as a mythical story that tries to provide answers to questions of meaning, but applies the criteria of a scientific theory or historiography to the Bible. With regard to explanations of the world (esoteric vs. scientific approach to the world; theory of evolution vs. belief in creation), the family conceive of science as superior, while viewing religion and Biblical texts as irrational and contradictory.

*5.2. Ethics: Not Founded on Religion*

There is agreement in the family that it is difficult to acquire a religious faith without religious socialization.[24] However, with the exception of the grandmother, they do not see the lack of faith as a deficiency; rather, the scientific approach to the world is deemed satisfactory and sufficient, and replaces faith.[25] At the same time, religiosity and faith are fundamentally respected on a personal level, which is also expressed in the fact that the children in all generations are free to be baptized and confirmed.[26] While the older daughter says that she lacks knowledge and argues that knowledge about religion is part of general education, the parents welcome the fact that their daughters engage with religion and ethical questions without demanding this of them: "it also goes into other areas then, like these ethical areas of euthanasia, abortion, stem cells, and I mean you have also just done a bit of that at school" (G2_F: 427–429). When it comes to ethics, the father draws a sharp line with regard to the church and theology, where they claim sovereignty of interpretation: "I personally ... not only cannot understand it, but I also find it wrong that the church is always automatically given a say in society" (G2_F: 431–434). From his perspective, this hinders rather than promotes an "ethical discussion", since it gives religion too much importance in ethical questions, which makes him feel unrepresented in society. This demarcation thus concerns the status of religion and church on the societal level. G2_F also rejects the atheist Gregor Gysi's statement that he cannot imagine a society without religion because he deems religion to be the "cement of society" (449), arguing instead that "religion influences me personally and by that I mean me and my family in a way that I find absolutely unacceptable" (452–453). The family feels strongly committed to

the common good and therefore also criticizes religion if it prevents rather than promotes engagement: the older daughter mentions posts calling for prayer after the attacks in Paris (2015), which she finds "strange" because "for me there is no point in praying—you should instead donate or help" (G3_D1: 484–485). There is unanimity in the family here: it is a matter of actively helping and being good to others, while the requirement to pray only serves to calm one's own conscience.

Taking an ethical stance that prioritizes the common good, the members of the family criticize religion as inhumane when it comes, for example, to questions of homosexuality, abortion, and suicide,[27] which they regard as personal decisions. In their drawing of boundaries, though, they distinguish between personal faith, which they respect, and the societal level, where they wish to deprive religion of any justification. Because of his rational approach to the world, the father looks for secular reasons why something happens or someone makes a decision, and rejects religious interpretations of life. He rejects "Christian values" if they mean that people are denied the freedom to make an independent decision. The following sharp secular statement most clearly shows his non-religious habitus, citing above all the debate on abortion and taking a position that clearly opposes religious evaluations: "when I have the feeling that this has been discussed by ethicists, legal experts, doctors, women and so on, but not by people who feel that life is something divine ... , I think it is no problem to abort a zygote, a fused sperm and egg cell, there is nothing divine about it, no one feels pain" (G2_F: 1211–1215). However, he stresses that, while the embryo may not feel pain, the pregnant woman certainly does, since she is in the situation of having to make this decision personally. He draws a strict line when it comes to religion having an influence with regard to a personal decision.

## 6. Conclusions

The aim of this paper was to show in detail how a non-religious habitus and a secular relationship to the world developed from lived values and their transmission in a family over several generations. Concluding, I will summarize the societal and familial conditions under which such a secular habitus can emerge. I will then argue that this habitus is not primarily based on demarcation from religion, but boundaries are drawn when religion interferes with secular interests. Following Quack and Schuh's distinction between "indifference to religiosity" and "indifference to religion" (Quack and Schuh 2017, p. 3), we can clearly identify indifference to religiosity but not towards religion.

It is true for all three generations that they live in a social context where non-religiosity is accepted: while for the early GDR a religious commitment was rather a disadvantage, in the FRG a decline in church religiosity slowly started in the 1960s and society pushed toward liberalization, secularization, and pluralization (Gärtner 2019a; McLeod 2007; Pollack and Rosta 2017). Under these conditions, both grandparents shared a non-religious childhood in the GDR and did not take up religious practice later. While the grandfather received a scientific education in his phase of adolescence in the GDR, the grandmother already spent her phase of adolescence in the FRG, got confirmed, and retains a kind of—as she puts it—"childlike faith", but she does not identify it as religion herself or pass it on. According to the reflections of the members of the family, a positive scientific stance, reliability, and security in the family enable them to grow up without religion or the need of a higher power.

Thus, religion is absent from the family's everyday life, and at most plays a situational role. Their mostly secular environment, where problems are both perceived and solved from a secular perspective, gives rise to the development of a primarily secular relationship to the world. The absence of religiosity or belief is not experienced as a deficiency but is based on a positive relation to science and reason as a point of reference for their own non-religious identity (Wohlrab-Sahr and Kaden 2013, p. 197). On this basis, the family separates private and public spheres: while they recognize and respect religiosity as a private phenomenon (this also applies to the personal faith of family members), they deny religion any right to interfere at the societal level of the public sphere. Against this

background, contact with religion in certain situations does not pose a problem. It is only when religious interpretations interfere with the secular interpretation of the world, or when their own way of life is impaired or threatened by religious values, that they draw boundaries.

The leading demarcation is that between science and religion. These realms are not understood as two separate and autonomous spheres; rather, the family tends to place both spheres in a hierarchical relationship, with science seen as superior to religion. Accordingly, demarcations become relevant when religion is interfering with the family members' secular interests on areas like politics, ethics, and science. These are based on the claim to a rational explanation of the world and to their own relation to values, one based on an ethics that is not founded on religion. Basic ethical attitudes and values draw on scientific humanism and primarily concern decisions regarding a person's right to determine her own life in issues such as abortion and euthanasia, but also intolerance of sexual orientations, e.g., homosexuality. Interference in private decisions based on a religious morality that conceives of life as divinely bestowed and thus as not belonging to the individual herself triggers a conflict that provokes sharp demarcations. They criticize in particular the church as a public actor and argue that the church has no right to exert an influence on legislation.

**Funding:** This research received an external funding from the John Templeton Foundation (#61361).

**Conflicts of Interest:** The author declares no conflict of interest.

## Notes

1    My data are drawn from a family interview conducted as part of the research project "The transmission of religion across generations: a comparative international study of continuities and discontinuities in family socialization", which focuses on the transmission of religion, faith, beliefs, and values. The project is funded by the John Templeton Foundation, which also provided a grant to enable the publication of this article. The opinions expressed here are those of the author and do not necessarily reflect the views of the Foundation.

2    I would like to thank Johannes Quack for pointing this difference out to me. I would also like to thank the reviewers for their constructive and helpful comments.

3    In contrast to "atheism", which develops in opposition to "theism" (Bullivant 2012), "scientific atheism" is a communist ideology with a scientific underpinning (Schmidt-Lux 2008). In West Germany, on the other hand, a *secularism* developed in the 1960s that, also inspired by Marxism and being ideological in nature, had definitely emancipatory and enlightenment intentions (Franzmann et al. 2006, p. 11).

4    As a result of people leaving the church in the last decade, around 27% of the total population still belonged to the Catholic church in 2020, just under 25% to the Protestant church, and almost 41% had no religion. Available online: https://fowid.de/meldung/religionszugehoerigkeiten-2020 (accessed on 24 February 2022).

5    On religiosity in East Germany, see the contribution by Olaf Müller and Chiara Porada in this volume.

6    It is not only international quantitative studies (such as WVS; EVS; Bertelsmann Religion Monitor) that show this decline due to intergenerational change (Voas and Doebler 2011; Pollack and Müller 2013, p. 15; Müller 2013, p. 220), but also our own quantitative survey. In their contribution in this volume, Olaf Müller and Chiara Porada demonstrate the decline in religious upbringing in families across family cohorts in Eastern Germany. Bullivant also shows for children of atheist or non-religious parents in the US that their identity differs from that of their parents' generation: "The parents had to hide their non-religion, which still causes problems for them, but their children are just starting to learn how they can be openly non-religious without massive discrimination" (Bullivant 2019, p. 100). This generational change is in line with the normative American identity of "tolerance and individualism", which leads them to claim "acceptance and validity for their non-religious identities in the American social, cultural, and religious environment" (Bullivant 2019, p. 97).

7    The relatively large timespan of the birth cohorts within the different generations is due to the fact that we have one four-generation family in our sample, where G3 are already parents themselves, and in another family, it is the second marriage for G2, with the husband being clearly older than his wife.

8    The interview was conducted in December 2019. G1–G3 = 1st–3rd generation; GM = grandmother; GF = grandfather; F = father; M = mother; D1 = daughter 1; D2 = daughter 2.

9    On the religious background of and "forced secularization" in the GDR in the 1950s, see Pollack (2009) and Wohlrab-Sahr et al. (2009).

10   The grandfather recounts that he received a scientific education, "which ends for the young people in this youth initiation ... that was actually, a bit like an approach to, also an attempt to get some kind of belief" (G1_GF: 259–262).

[11] G1_GF: "the sister's death was not a big sad event for the other sister; she kind of thought that the sister would finally find release ... in heaven and that she would be better off there than here; that was sort of impossible to understand that someone could see death so differently, but that was actually the only thing that moved me a bit, that if you have such a strong faith, maybe you can get over certain things more easily" (291–297).

[12] G1_GM: "there must also be something there, well, charity and gratitude, I actually find that quite important, I don't know if it's important anywhere in the Bible or, well, maybe with Jesus, he was so humble" (588–591).

[13] In German, we would use the term "Unverfügbarkeit", whose literal translation "unavailability" is not entirely adequate. *Unverfügbarkeit* is about the awareness that life is not wholly subject to our control and thus the acknowledgment of its limits. It leads to a certain humility—and, in the case of the grandmother, to an attitude of gratitude. The moment of the "*Unverfügbaren*" represents a central characteristic in the concept of resonance, which Hartmut Rosa understands as a counter-concept to "*Entfremdung* (alienation)" (Rosa 2017, p. 38).

[14] I reproduce the quotations from the family members in their oral form. I am aware that this makes it difficult to read because oral language does not conform to written grammar. Omissions are marked by dots.

[15] The further course of the interview shows that the great-grandfather's religiosity did not convince anyone in the family, as his churchgoing apparently did not lead to "respectful interaction" or a reconciled attitude in life.

[16] Since he did not report any experience of difference before, we assume that, like his peers, he took part in religious lessons at school.

[17] This general tolerance of religion may be explained by the fact that neither he nor his parents experienced a conflictual detachment from religion. Bullivant (2019) shows comparable patterns for the non-religious youth in the US context.

[18] She narrates that her father was an atheist (G1_GM: 1020).

[19] G1_GM: "you did realize that there is something, and that's how it has remained" (311).

[20] Inglehart (2000) attributes the fact that the young generation is strongly oriented towards family and relationships to their growing appreciation of post-materialistic values, this being especially the case among those who have experienced economic and physical security as a matter of course.

[21] G3_D1: "We should really mention that the school has a choir and then you are in this church and sometimes the whole school comes into the church, the light is turned off, the choir comes in with candles [G3_D2: yes], that's just a great feeling when then, the singing echoes in, this building, and that's, but also at this school in England, it was that you always met in the morning and [G3_D2: yes] you always prayed (?) [G3_D2: not always, but usually], but sometimes you prayed [G3_D2: yes], and then everyone sat together, I don't know, 400, 500 pupils [G3_D2: about 400]" (148–155).

[22] While their father got confirmed because he followed the common practice of his peers, the daughters' decision against confirmation required a greater personal justification, as it differs from that of their peers (see also Quack 2017, p. 203).

[23] Lois Lee also reports interviewees with positive non-religious identities who are highly critical of people who have had a church wedding or "used church services when not actively religious, viewing it as hypocritical and morally weak" (Lee 2014, p. 474).

[24] G2_F: "if you don't bring this spark in at the beginning, then it will be difficult later" (394–395).

[25] G2_F: "I don't believe in anything and I don't need to, either" (442–443)—he reinforces this attitude with regard to his own death: "I think Peter Ustinov once said that in a plane that's about to crash everyone becomes a believer; that's an amusing anecdote, but I would disagree completely ... I would probably be terrified, but in the last two seconds would think of my family" (443–446).

[26] The father says: "my wife and I would have no problem at all if you said you wanted to be baptized" (388–389), to which the eldest daughter replies: "yes, I know that too . . . , that you think it's totally OK and that if we want to do it we could" (G3_D1: 390–391).

[27] The grandmother, who was a school secretary, tells of a teachers' meeting where suicide was discussed: "all of them are Catholic, five Catholic teachers, and they started to talk about the fact that I had said, for some reason, well, God or Jesus would forgive someone who is in such a difficult situation and commits suicide, ... there was ... horror at the table, it's impossible that someone takes their own life ... and then we got into a big discussion, and then I thought . . . that can't be Jesus or God that he condemns a person who is in such a bad place" (G1_GM: 524–530).

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
