# Peer review of "Secularity as a Point of Reference: Specific Features of a Non-Religious and Secularized Worldview in a Family across Three Generations"

_religions, doi:10.3390/rel13060477_

Round 1
Reviewer 1 Report
Dear author,
I have enjoyed reading your family interview and situating the lived experiences of nonreligion and when and if religion poses a problem to their worldviews.
Please find my comments below which I hope are helpful:
Line 10: In the abstract, you refer to "her view of religion" – it's not clear at this stage if this is about gender and non-religion and who "her" is. So it might be worth just some brief clarity here on the sample/method.
Line 52: It might be worth just unpacking the term "scientific atheism", its links with communist ideology and how this differs from "atheism". And how this communist ideology divided the country regarding religion/nonreligion.
Line 66-84: Some more recent figures on generational changes in religion would be helpful here. I would recommend reviewing data from somewhere like the European Social Survey (2016) which shows that when German adults are asked about religious affiliation and belonging to any denomination 39% say none and this is higher for under '30s (44% say none). I think something more recent like this would be helpful to strengthen the argument of generational differences in German religious belonging.
Line 96-103: There needs to be more of a discussion on the method here, particularly the sample and why this family was selected. How were they recruited? And also how the data was analysed. Were the participants all interviewed at the same time (this isn't clear when reading lines 157-161)? What questions were asked? Please expand this section.
Line 189: A very small point but why "apparently" is his recounting of events not credible? This also speaks a little to the need to develop the methods section beyond just who was interviewed.
Line 287: An interesting point, but references could be added in to strengthen. Pippa Norris and Ronald Inglehart's work on Sacred and Secular (2011) might be helpful to explain this when a society/persons are comfortably and there are reduced threats to survival, religion may be seen as less important.
Line 310-311: You make the point that the granddaughters have friends who have been baptised but who they suspect do not believe in god and then go on to say how this demonstrates a weak Christian faith. I would recommend further bolstering your literature review earlier on with some ESS or EVS data on younger people and increased secularity/religious rituals to help make this point which you could return to here. Right now this is more anecdotal of what they suspect to be true.
Recommended additional reference:
I would expect to see Lois Lee's (2014) Secular or nonreligious? Investigating and interpreting generic ‘not religious’ categories and populations. Religion. Vol 44. No 3. 466-482 discussed alongside Bullivant. Lee's article engages with Indifferentism (p.474). This should be used alongside Quack and Schuh's argument.
Reviewer 2 Report
There are several style and minor grammatical adjustments needed throughout - several items are highlighted in blue but without any consistency throughout the article. This might be an editorial decision, but it's distracting for the reader. At points there are some slight grammatical issues (e.g. I would change Traditionally to Historically in the first sentence) throughout but they are minor and can be addressed easily.
